# Geometric-based approach for linking various building measurement data to a 3D city model

**Yoshiki Ogawa** [1]*, **Go Sato**[2], **Yoshihide Sekimoto**[1]

**1** Center for Spatial Information Science, the University of Tokyo, Tokyo, Japan, **2** Department of Civil Engineering, the University of Tokyo, Tokyo, Japan

* ogawa@csis.u-tokyo.ac.jp

**Data Availability Statement:** The image data and LP/ MMS data analyzed in the study are provided by the Geospatial Information Authority of Japan (https://www.gsi.go.jp/ENGLISH/index.html) and Shizuoka prefecture via Association for Promotion

## Abstract

Currently, the Ministry of Land, Infrastructure, Transport, and Tourism (Japan) is in the process of developing an open 3D city model known as PLATEAU. Abundant measurement data related to buildings, including maps produced by private companies and mobile mapping system point clouds, have been collected to enhance the value of the 3D city model. To achieve this, it is necessary to identify the buildings for which measurement data is available. In this study, we propose and evaluate an efficient matching method for various building measurement data, primarily using geometric properties. In Numazu city, PLATEAU IDs were assigned to 88,525 Zenrin buildings as part of a private map. The results indicate that 90.6% of the polygons were matched. For aerial images, 93.6% of the extracted buildings matched the PLATEAU buildings, although only 70.9% of the PLATEAU data was extracted from the images. Using the level of detail 1 and 2 models, 46 textured building files were created from the mobile mapping system point cloud. In addition, the cover ratio for the laser profiling point cloud was mostly greater than 40%, which was higher than that of the mobile mapping system.

## 1. Introduction

In recent years, numerous countries around the world have developed 3D city models. Wide area semantic-based 3D city models have been developed in 21 cities in nine countries worldwide [1] and the PLATEAU project aims to develop 3D city models as open data for cities throughout Japan [2]. These models are expected to be beneficial for urban activity monitoring, disaster prevention, city planning, and more. The value of these models can be further enhanced by incorporating information on buildings, such as textures and building use.

This study focuses on three types of information used to augment building data: 2D footprint data, aerial photographs, and 3D point clouds. In Japan, a private company produces 2D footprints, also known as residential maps, based on field surveys, measurements, and drawings, which include information on building names and uses. Aerial images, which contain information on the time-series changes of buildings and ceiling surfaces, are provided by the Geospatial Information Authority of Japan (GSI) and private companies. 3D point clouds, containing information on the color and shape of buildings, are measured by aerial laser surveying or specialized vehicles called mobile mapping system (MMS). Because these data were

of Infrastructure Geospatial Information Distribution (https://front.geospatial.jp/). These are open data and can be shared publicly. Our code is shared via GitHub (https://github.com/Project-PLATEAU/UC22-008-Building-matching-WebAPI).

**Funding:** The author(s) received no specific funding for this work.

**Competing interests:** The authors have declared that no competing interests exist.

created independently from the 3D city model, it was necessary to link them to the 3D city model.

This study presents numerous technical advancements. Previous studies have evaluated 2D footprint similarity by utilizing the distance between representative points and overlapping areas to merge multiple Geographical Information System (GIS) databases that do not share the same geometry [3, 4]. These studies commonly cited the difficulty of matching one-to-many polygons as an issue. Furthermore, a study created a 3D buffer from the corresponding wall elements and extracted the point cloud [5] in the matching of 3D point clouds to models, but automating the entire point cloud matching process, including generating a 3D city model with wall textures and providing a WebAPI service, has yet to be achieved. Thus, it is essential to develop a WebAPI to facilitate the open utilization of the matching algorithm and to meet social needs and technological advancements.

Considering social needs and technical advancements, the research question of this study is to systematically organize an automatic linking method that can link various types of exterior measurement data, which are voluminous and diverse, to a 3D urban model.

To effectively utilize the three types of building measurement data (2D footprint, aerial images, and 3D point clouds) from various sources, we developed a matching method for reliable 3D building models (PLATEAU) and implemented it as a Web API. The matching was performed using three typical examples: residential maps, aerial images, and 3D point clouds (Fig 1).

The remainder of this paper is organized as follows: Chapter 2 describes existing technologies and previous studies related to this study. Section 3 describes the datasets used in this study, and Section 4 describes the methodology used. Section 5 presents the results and discussion of the study, and Section 6 concludes this thesis and presents future research directions.

The WebAPI developed in this study is accessible at the following URL: https://github.com/Project-PLATEAU/UC22-008-Building-matching-WebAPI.

## 2. Related work

### 2.1 Wide area 3D city model

The development of wide-area 3D city models is being promoted in numerous countries worldwide. This study is based on Geography Markup Language (CityGML) 2.0, which defines the level of detail (LOD) concept for the representation of geographic features, such as buildings [6]. The definition of LOD in CityGML 2.0 is as follows: LOD1 defines a building shape by a series of prisms and does not represent the outer façade, whereas LOD2 represents detailed roof shapes and outer façades. For instance, the shape of an arcade in a shopping district was added to LOD2. Building openings, such as doors and windows, are added in LOD3, and the interior of the building, including rooms and interior installations such as desks and chairs, is represented in LOD4.

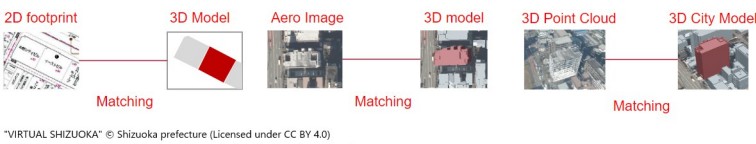

"VIRTUAL SHIZUOKA" © Shizuoka prefecture (Licensed under CC BY 4.0)
Aerial photograph ©Geospatial Information Authority of Japan (Licensed under CC BY 4.0)

**Fig 1. Conceptual illustrations for the three types of matching targeted in this study.**

The 3DCityDB, a software package for handling CityGML, lists 21 cities in nine countries worldwide for which wide-area CityGML 3D models have been developed, including Helsinki, Singapore, and Tokyo [1]. The Helsinki 3D, a semantic-based model in the CityGML format, is an example of 3D city model development [7]. The model was utilized for comparative studies of urban energy consumption and the visualization of future development models. Another example of wide area 3D city models is "Virtual Singapore" [8], which covers the entire area of Singapore and employs the CityGML data format. In Japan, Project PLATEAU, a 3D city model in the CityGML format, is being developed, and approximately 200 cities, including Tokyo, are targeted for development by FY2023 [2]. The target number of cities to be developed is set at approximately 500 by FY2027, corresponding to approximately 30% of the total number of municipalities in Japan.

The integration of diverse datasets is critical in creating wide-area 3D city models. Seto et al. 2020 [9] successfully integrated and visualized complex and diverse datasets obtained in their study area, including aerial images, 3D building models, 3D point clouds, and data on people flow and traffic, while reducing data volume. However, data development for 3D point clouds is currently limited to a single location. In Sofia, Bulgaria, Dimitrov and Petrova-Antonova 2021 [10] integrated DSM/DEM (TIFF files), residential map data (point format), and building data (2D polygons). The study successfully created an LOD1 building model in the CityGML format without being hindered owing to topography.

## 2.2 2D footprint matching

Previous studies that integrated 2D footprint data include Tong et al. 2009 [3] and Ruiz-Lendínez et al. 2017 [4]. Tong et al. 2009 [3] employed polygons extracted from paper maps and QuickBird images as the GIS polygons, whose shapes did not match. Polygon similarity was assessed based on the distance between the representative points and the overlapping area. Similarly, Ruiz-Lendínez et al. 2017 [4] used data created by the Spanish National Geographic Institute (BCN25) and the Andalusian Institute of Statistical Mapping (MTA10) as two GIS polygons whose shapes did not match. They used a weighted linear combination of polygon similarity values obtained from seven criteria: the number of convex vertices, number of concave vertices, perimeter length, area, minimum second moment, Arkin Graph Area, and minimum bounding rectangle. Both studies cited the difficulty in handling one-to-many matching as a challenge in 2D polygon matching.

A tree structure is effective for locating 2D polygons in close proximity to each other. Creating an R-tree using PostGIS, a component of 3DCityDB, accelerates the spatial retrieval of 2D polygons [11]. Quadtree-based partitioning, such as Open Location Code [12], is a simpler spatial partitioning method compared to R-tree. Quadtrees possess advantages such as low overhead and applicability to distributed databases [13, 14].

Converting aerial images into 2D polygons is an instance segmentation problem in image recognition. State-of-the-art models for instance segmentation include Mask R-CNN [15], MS R-CNN [16], and RefineMask [17]. Super-resolution techniques are beneficial for object detection in aerial images. State-of-the-art models for super-resolution, such as SRGAN [18], ESRGAN [19], and SwinIR [20], are based on GAN techniques. In Japan, the Geospatial Information Authority of Japan (GSI) provided an open-source satellite image dataset, which Chen et al. 2023 [21] utilized to study the extraction of building shapes. However, the GSI dataset has some drawbacks, such as color differences and low resolution, which were addressed by enhancing the Mask R-CNN, color adjustment, and super-resolution techniques by extending ESRGAN. Because the extracted building geometry is independent of the existing geographic information database, a matching technique between 2D polygons is required to

use the building information extracted using this method in combination with existing geographic information.

## 2.3 3D point cloud matching

Studies linking airborne laser scanner (ALS) point clouds to LOD1 models for mapping 3D point clouds to 3D urban models exist. Park and Guldmann 2019 [22] estimated building heights for the LOD1+ model (an extended white box model in which one building is represented by a combination of white boxes with multiple heights) by extracting only the point cloud of building roof surfaces from the ALS point cloud. Albeaik et al. 2017 [23] corrected a low-resolution noisy ALS point cloud to create an LOD1 building 3D model.

Building texture images based on airborne data were created in the following studies: Lee and Yang 2019 [24] utilized oblique images captured from aircraft to generate textured images of the sides of buildings to improve the quality of 3D urban models. Some studies that utilized oblique images to create 3D city models employed the structure from motion technique [25].

Several studies have created building texture images based on data from MMS. Yang 2019 [26] created a building texture image by developing an image acquisition and distortion correction system using MMS. Kelly et al. 2017 [27] and Femiani et al. 2018 [28] utilized Google Street View (GSV) images [29] to generate building-side images. Kelly et al. 2017 [27] extracted individual buildings from MMS images using edge scores, determined the representative colors of buildings using the most frequent values, and estimated the locations of structures, such as windows and doors, to produce a textured 3D city model for 1011 buildings in 37 blocks. Liu et al. 2020 [30] set up a loss function based on the symmetric rectangular shape of windows and doors for object detection in images of building sides. By contrast, Dai et al. 2021 [31] performed a study on images of suburban areas instead of urban areas. Femiani et al. 2018 [28] used angle information to crop buildings from GSV images. Tian and Wang 2023 [32] proposed a method to find the best match between the 2D photos captured at unknown distances and the actual object in the captured stereo images through perspective projection and model matching. The matching fitness function was optimized using a genetic algorithm. The experimental results were obtained using two different objects and the accuracy was found to be better for the industrially manufactured product (sunscreen) compared to the agricultural product (pear) with irregular shape variations. Scientific research addresses the thermal image mapping onto 3D models for visualization and analysis. Antón and Amaro-Mellado 2021 [33] developed an open-source software graphical method to produce 3D thermal data from infrared thermography (IRT) images for temperature visualization and subsequent analysis.

E. Oniga 2012 [34] proposed an algorithm for the semiautomatic texture generation based on color information, RGB values of every point captured by terrestrial laser scanning technology, and 3D surfaces defining building facades generated using commercial 3D software. The operator needed to define the limiting value, i.e., the minimum distance between a point and the closest surface. In this study, the threshold used to choose the points to be projected was set automatically. Beil et al. 2021 [5] coupled a 3D city model provided by CityGML with a point cloud. To link the MMS point cloud to CityGML, a 3D buffer was created from the corresponding CityGML elements, and the point cloud was extracted. However, Beil et al. 2021 [5] have not yet automated all point-cloud matching tasks, including the output of 3D city models with wall surface textures, and provided services such as WebAPI. Open3D [35] is an open-source software package that can be used for point-cloud extraction, and functions in Open3D can be used to select and save only points within a certain distance of the building in terms of the XY plane. However, using the functions included in Open3D, it is not possible to

extract a point cloud based on the distance between a point and a surface, considering the Z coordinate.

In this study, 3D city models with wall surface textures were made from open MMS 3D point cloud data using only free and open-source software rather than using commercial 3D software. The texture was added to the 3D city model provided by CityGML. All point-cloud matching tasks were automated and the service was provided as WebAPI.

## 3. Dataset

### 3.1 3D city model (PLATEAU)

The 3D city model data used in this study were the PLATEAU data provided by the Ministry of Land, Infrastructure, Transport, and Tourism. This study utilized LOD1 and LOD2 data from Numazu city, and the survey year for the PLATEAU data in Numazu city was 2015.

### 3.2 2D footprint (residential maps)

2D footprint data is the map data that has attributes related to the name and use of each building based on field surveys, measurement information, and surveys. In this study, we used 2D polygon data sold by a private company (Zenrin Corporation), mainly utilized in Japan. Data updates are performed once every year, and the data employed in this study were obtained in FY2021. The residential maps were used in this study as a 69.9 MB polygon file in GPKG format.

### 3.3 Aerial images

In this study, building footprints extracted from the orthorectified image dataset provided by the Geospatial Information Authority of Japan (GSI) in the method by Chen et al. 2023 [21] were matched to the 3D city model. The orthorectified image dataset is a distortion-free image of aerial photographs captured by the GSI, which can overlap with various types of geospatial information by adding precise location information. The aerial images of Numazu City, with a resolution of 20 cm, used in this study were captured in 2010 and 2012. For the matching process in this study, we used a 2D GPKG polygon file (138.8 MB), from which polygons were extracted from the orthorectified image dataset of Numazu City.

### 3.4 3D point cloud

3D point clouds are produced by aerial laser surveying or special vehicles called MMS and contain information on the color and shape of buildings. The 3D point cloud data files were downloaded from the G-Spatial Information Center website [36]. The point cloud data were generated in 2019, and the area to be downloaded was selected using a 400 m x 300 m mesh. The 3D point cloud utilized in this study pertains to a single mesh near the northern exit of Numazu Station, and both the data (201.4 MB) obtained by an aerial laser survey (LP) and the data (3.2 GB) obtained by a mobile measuring vehicle (MMS) were employed.

## 4. Methodology

### 4.1 Processes implemented in this study

In this study, we implemented a 3D city model building ID matching and suggestion process for 2D footprints and a wall ID assignment process for each point that comprises a 3D point cloud. Fig 2 depicts the overall processing of 2D footprints and 3D point clouds.

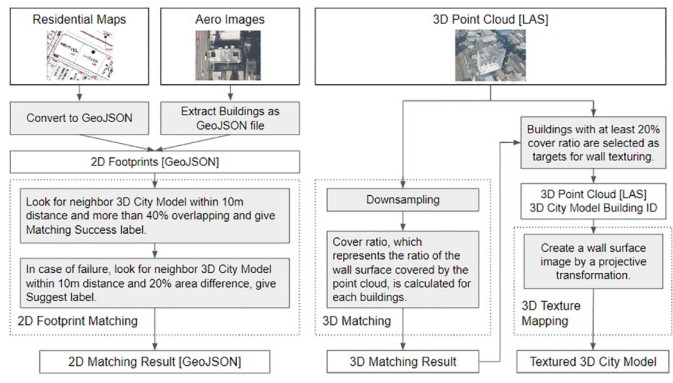

**Fig 2. Overall processing of 2D footprints and 3D point clouds.**

## 4.2 2D footprint matching

The matching results of the 2D footprints to the 3D city model were obtained based on the criterion that the distance between the centers of gravity should be within 10 m and the overlap ratio should be 40% or more. The overlap ratio is the intersection area of two polygons divided by the area of the smaller polygon. This criterion was set to enable the correct assignment or rejection of IDs for all data for one city block near the north exit of Numazu Station. By using this city block, it was confirmed that this criterion can handle the cases of one-to-many or one-to-none matching, which was considered difficult in previous studies. If no matching building is found in the aforementioned process, a building ID is suggested for a building with an area difference of 20% or less from the 2D polygons and located within 10 m of the center of gravity. If more than one matching buildings is found, the nearest one is chosen by comparing the distance between the centers of gravity.

Fig 3 presents conceptual illustrations of the matching based on this criterion. In the figure, (a1) shows an example in which the intersecting area exceeds 40% of the input 2D building polygons, and (a2) shows an example in which the intersecting area exceeds 40% of the building database polygons. By defining the overlap ratio as the intersection area of the two polygons divided by the area of the smaller polygon, matching was successful even when one polygon was larger than the other, as shown in (a2). Cases, where matching was not performed using this criterion, include (b1), where the polygons do not intersect and their centers of

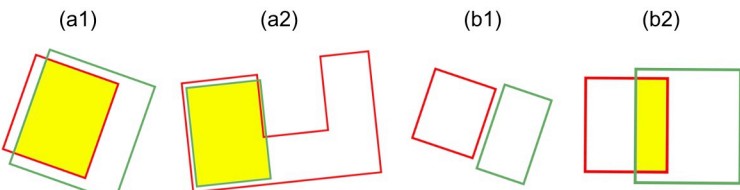

**Fig 3. Conceptual illustrations of matching based on distance and overlap ratio.** Red denotes input 2D building polygons, green denotes building DB polygons, and yellow represents intersections. [a1] An example of matching where the intersection exceeds 40% of the input 2D building polygon. [a2] An example where the intersection exceeds 40% of the building DB polygons and matches. [b1] Not matched because neither polygon intersects, and the center of gravity is extremely far. [b2] Not matched because the intersection area was less than 40% of the polygons in both polygons.

gravity are far apart, and (b2), where the area of intersection is less than 40% of the area of either of the two polygons.

## 4.3 3D point cloud matching

The cover ratio is calculated as the number of points close to the wall surface divided by the total surface area of the building, representing the proportion of the wall surface covered by the point cloud. After 1 m³ voxel sampling, the number of points within 1 m of the wall was computed. Buildings with high-quality wall images that met the condition of a cover ratio of at least 20% were chosen as targets for wall texturing. Points close to the wall surface were defined as those within 1 m of the wall surface, considering the sampling interval. A projective transformation of the wall surface was performed to generate a wall surface image from a 3D point cloud. Fig 4 illustrates an example of a point cloud downsampled in cubic units of 1 m per side and nearby buildings.

Fig 5 presents the creation of a wall surface image from a 3D point cloud. To create the wall surface image from the 3D point cloud, we first gathered points near the wall surface in the xz plane by rotating around the z-axis, projecting to the xz plane, and interpolating to the x and z coordinates of the wall surface using nearest-neighbor interpolation. In LOD2, many nonvertical wall surfaces were present. However, by estimating the normal vector of the surface from the three points that were not on the same line and constituted the LOD2 wall surface, the plane to be projected onto was determined, corresponding to the xz plane in LOD1. The wall texture was created by performing the same rotational shift, projective transformation, and nearest-neighbor interpolation as in LOD1. Initially, the resolution of the wall image was set such that 1 cm corresponded to one pixel. When the long side length exceeded 512 cm, the longer side length and width were set to 512 pixels. The areas without points are shown in gray color.

The imaging function collects all the points in the vicinity of the building and chooses the points to be projected based on the texture mapping method option. When using "- all: all points are projected," all points near the building are used. When using "- nearest: the closest point to each surface is mapped," only the point nearest to the surface to be projected is used.

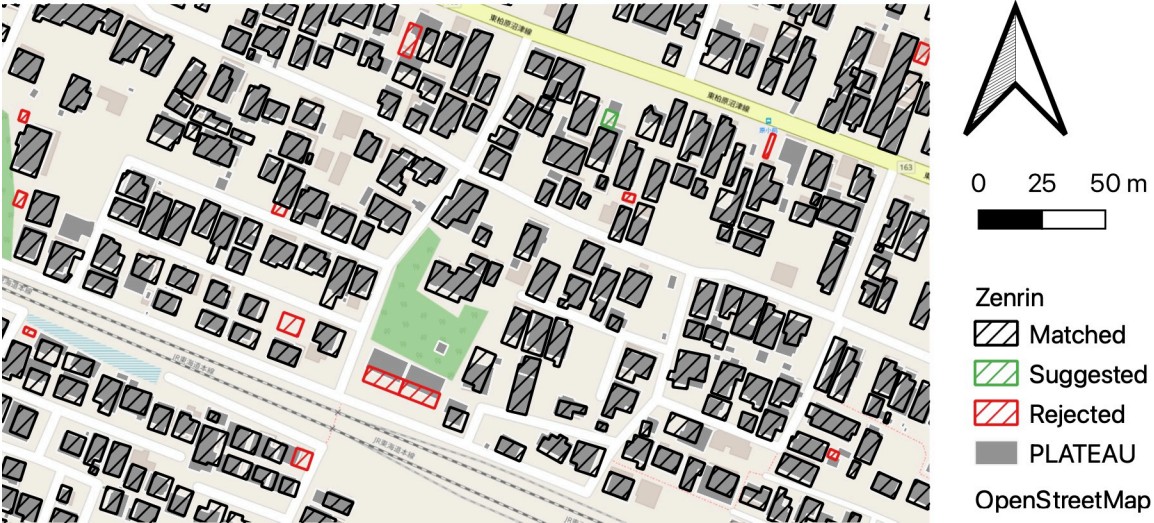

**Fig 4. Example of point cloud downsampled in cubic units of 1 m per side and nearby buildings.**

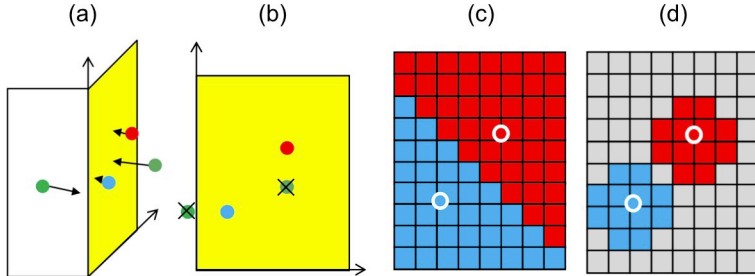

**Fig 5. Method for creating wall surface images from 3D point clouds.** [a] Projection transformation of a group of points on the wall surface (yellow) to create a textured image. [b] Delete points projected outside the surface and points whose distance from the surface is greater than the threshold value (*). [c] Nearest-neighbor interpolation. [d] Areas with no points are shown in gray.(*) The threshold is defined as the maximum distance between the wall and points inside the building that have the designated wall ID as the closest wall ID, preventing the wall from being imaged by the backside point cloud.

When using "- smart: Auto-detect maximum depth," a threshold is set as the maximum distance between the wall and points inside the building that have the designated wall ID as the closest wall ID. Points closer to the wall than the threshold are selected as the mapping targets.

## 4.4 Integrated execution of matching in WebAPI

The WebAPI was constructed on an AWS EC2 r5a.large instance (with 16 GB memory, 2vCPUs, and a 2.5 GHz clock frequency). All three matching methods were executed, and their execution times were measured in the WebAPI environment. The WebAPI was designed to enable both command line interface (CUI) and graphical user interface access to the matching methods.

The created Web API consists of three endpoints, as listed in Table 1. The first endpoint is 2D footprint matching, which sends a query for 2D footprint data expressed in GeoJSON polygons and returns the corresponding 3D city model building data in GeoJSON format. Experimental results for 2D footprint matching were generated from this endpoint. The second endpoint is 3D point cloud matching. When a 3D point cloud data file in LAS format is sent as

**Table 1. Created WebAPI endpoints.**

|  | **2D footprint matching** | **3D point cloud matching** | **3D texture mapping** |
|---|---|---|---|
| Function | When 2D building data expressed in GeoJSON polygons is sent as a query, the corresponding 3D city model building ID is returned in GeoJSON format. | When 3D point cloud data is sent as query data, a list of building IDs of the 3D city model corresponding to the point cloud and information necessary for coverage are returned in 2D GeoJSON format. | When 3D point cloud data and the building ID of a 3D point cloud city model are sent as query data, a 3D city model with the texture generated from the point cloud data mapped to the building is returned in Wavefront OBJ format. |
| Endpoint | /api/building2d | /api/pointcloud3d | /api/mapping3d |
| HTTP method | POST | POST | POST |
| Request Body | 2D building data expressed in GeoJSON polygons | Multipart form data, including 3D point cloud LAS files | Multipart form data, including 3D point cloud LAS files |
| Response format | GeoJSON FeatureCollection | GeoJSON FeatureCollection | Zip file |
| Main information contained in the response | 3D city model building ID and confidence information (matching or suggestion) | Building ID of the 3D city model, information on which the cover ratio calculation is based (building surface area and number of points near the wall surface) | Zip file that becomes a textured 3D city model in Wavefront OBJ format when unzipped |

query data, a list of building IDs of the 3D city model corresponding to the point cloud and information necessary for cover ratio calculation are returned in 2D GeoJSON format. Experimental results for the cover ratio calculation were generated from this endpoint. The third endpoint is 3D texture mapping, which sends the 3D point cloud data and the ID of the building data of the 3D city model as query data and returns the 3D city model data with the texture generated from the point cloud data mapped to the building as a Wavefront OBJ format file. Experimental results on textured 3D city models were generated from this endpoint.

Further explanation on the WebAPI is accessible at the following URL: https://project-plateau.github.io/UC22-008-Building-matching-WebAPI/.

## 5. Results and discussion

### 5.1 2D footprint matching

The results of 2D footprints are listed in Table 2. Using the distance between the centers of gravity and overlap ratio, we matched building IDs for 90.6% of the footprints, rejected 8.2% (no matching), and classified 1.2% as footprints requiring visual judgment. In Ruiz-Lendínez et al. 2017 [4], 90.6% of the matches were smaller than the value of 97.9%. However, considering the differences in datasets used and the fact that the nonmatching footprints include successful cases of one-to-none matching, i.e., correct rejects, the method employed in this study is more accurate. Despite the simplicity and small number of features used, no significant difference existed in accuracy compared to Ruiz-Lendínez et al. 2017 [4].

The matching and suggestion results for the residential map polygons are shown in Fig 6, depicting all results within the ranges shown. Notably, the suggestion in the figure is for a 3D city model (PLATEAU) polygon that exhibits similar shapes in the northeast direction but is not matched owing to a low overlap ratio, highlighting the importance of incorporating a suggestion function.

The results for aerial image processing are listed in Table 3. Although the number of buildings extracted from aerial images was only 70.9% of the total number of polygons in the 3D city model (PLATEAU), 93.6% of the extracted buildings matched the PLATEAU building IDs, and 6.3% were successfully rejected (no matching). Fig 7 shows the process results, including matching and suggestion of polygons extracted from aerial photographs. The matching (black shaded line) and suggestion (green shaded line) results were all valid within the range shown in the figure, and the rejects corresponded to buildings that existed in aerial images but not in PLATEAU and buildings whose building outlines were misdetected. The former is effective in tracking time-series changes in buildings, whereas the latter is useful in detecting errors in the building contour extraction model.

**Table 2. Results for 2D footprints.**

|  | Whole Numazu city |
| --- | --- |
| Number of polygons included in the 3D city model (PLATEAU) | 113,691 |
| Number of 2D footprints | 88,525 |
| Number of 2D footprints in residential maps where building ID was successfully matched (percentage) | 80,239 (90.6%) |
| Number of 2D footprints in residential maps where building ID assignment was rejected (percentage) | 7,228 (8.2%) |
| Number of 2D footprints in residential maps where building ID was suggested and required visual confirmation (percentage) | 1,058 (1.2%) |

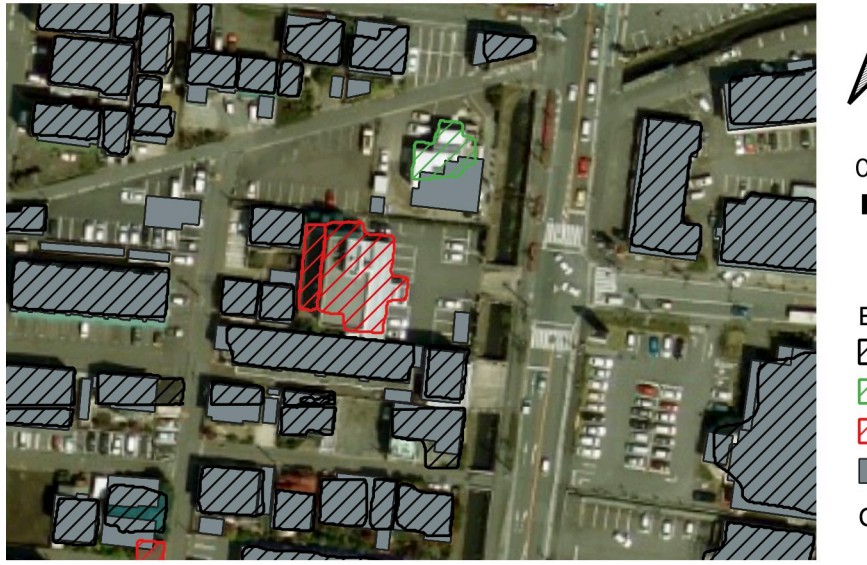
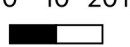

Aerial photograph ©Geospatial Information Authority of Japan (Licensed under CC BY 4.0)

**Fig 6. Results of the matching and suggestion process for 2D footprints.**

## 5.2 3D point cloud matching

Fig 8 shows the cover ratio calculation results for the MMS point cloud, whereas Fig 9 shows the cover ratio calculation results for the LP point cloud. The cover ratio of buildings located north-south near the center of the figure and close to the main street where the MMS traveled was generally over 20% for the MMS point cloud and over 40% for the LP point cloud measured from the sky. This suggests that the cover ratio difference was determined by the measurement method. Fig 10 shows the results of generating wall images for multiple buildings in the MMS point cloud, comparing the outcomes of wall image generation depending on the level of model detail. Comparing the LOD 1 and LOD 2 results, the LOD 2 building model more closely resembled the actual building, resulting in fewer gray pixels (no points) on the walls.

When generating wall images for LOD2 buildings, projecting only points with the corresponding wall ID onto the wall surface resulted in image generation issues owing to geometry overlap. For such buildings, the optimal results were obtained by projecting all building points

**Table 3. Results for aerial image processing.**

| | Whole Numazu city |
| --- | --- |
| Number of polygons included in the 3D city model (PLATEAU) | 113,691 |
| Total number of polygons extracted from aerial images (percentage compared to the number of polygons in the 3D city model) | 80,645 (70.9%) |
| Number of extracted polygons with successfully matched building IDs (percentage compared to the number of polygons extracted from aerial images) | 75,458 (93.6%) |
| Number of extracted polygons with successfully matched building IDs (percentage compared to the number of polygons extracted from aerial images) | 5,062 (9.3%) |
| Number of extracted polygons with suggested building IDs requiring visual confirmation (percentage compared to the number of polygons extracted from aerial images) | 125 (0.2%) |

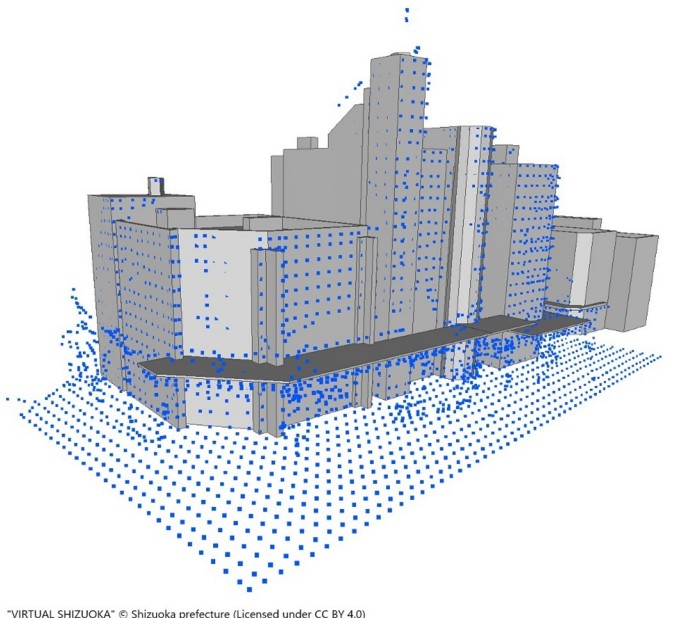

"VIRTUAL SHIZUOKA" © Shizuoka prefecture (Licensed under CC BY 4.0)

**Fig 7. Results of the matching and suggestion process for polygons extracted from aerial images.**

that were within the threshold distance from the surface. Fig 11 compares the texture changes for a building with a complex LOD2 geometry as a result of changes in the set of projected points, as well as the texture mapping results for an LOD2 building with different mapping settings. Even with the improved thresholded projection algorithm, projecting using the LP point cloud resulted in the majority of the wall surface being covered in gray owing to the small number of points.

## 5.3 Integrated execution of matching in WebAPI

Table 4 summarizes the matching results of various measurement data in the WebAPI. For the 2D footprints (residential map), 90.6% of the 88,525 polygons in Numazu city matched with

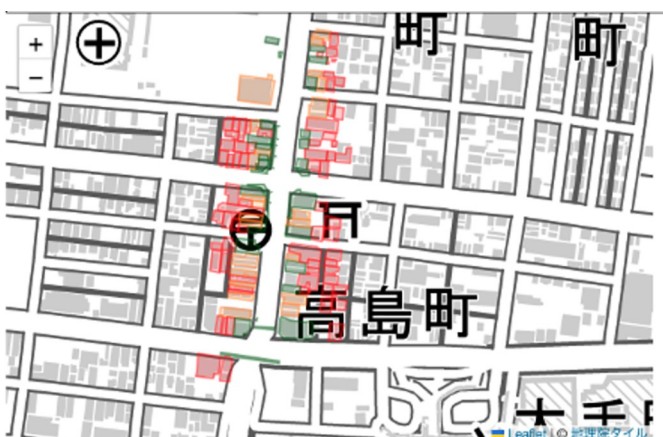

**Fig 8. Result of the cover ratio calculation for the MMS point cloud.** Green: Buildings with a cover ratio of 40% or more. Yellow: Buildings with a cover ratio between 20% and 40%. Red: Buildings with a cover ratio of less than 20%.

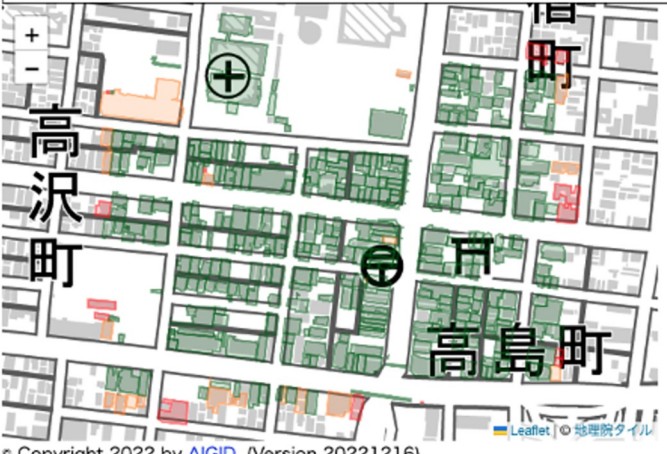

**Fig 9. Result of the cover ratio calculation for the LP point cloud.** Green: Buildings with a cover ratio of 40% or more. Yellow: Buildings with a cover ratio between 20% and 40%. Red: Buildings with less than 20% cover ratio.

building IDs, whereas 8.2% were rejected (no matching) and 1.2% required visual judgment (classified as gray). Limiting the visual judgment to 1.2% can significantly reduce the time required for confirmation in practical terms. The time required to match the entire city was 104 s, considered practical in terms of execution speed.

For the aerial photographs, only 70.9% of the PLATEAU buildings were extracted in the preprocessing step for matching. Out of the 80,645 extracted polygons, 93.6% had building IDs matched, whereas 6.3% were rejected (no matching) owing to various reasons such as

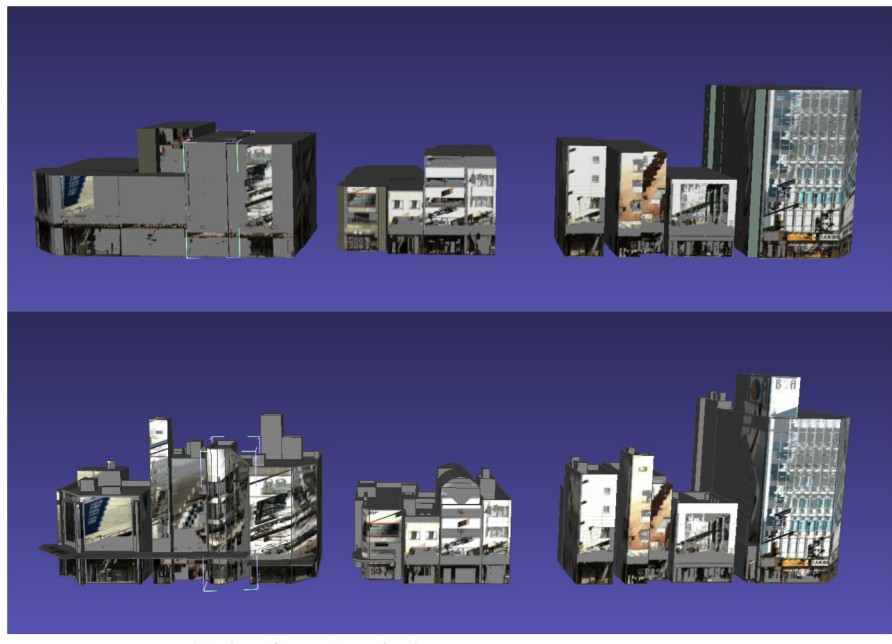

**Fig 10. Comparison of wall surface image generation results by level of detail.** (Upper) LOD1. (Lower) LOD2.

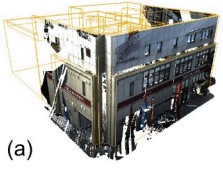 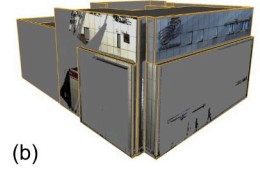 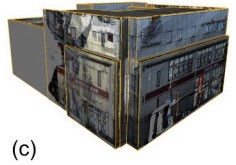 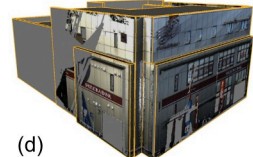

(a)　　　　　　　　(b)　　　　　　　　(c)　　　　　　　　(d)

**Fig 11. Texture mapping results for an LOD2 building with different mapping settings.** [a] LOD2 building model and MMS point cloud. [b] "all": All points were projected. [c] "nearest": The point closest to each surface is mapped. [d] "smart": Maximum depth is auto-detected.

incorrect extraction and new construction or demolition of buildings. Visual confirmation was required for both suggestions and rejections. The matching time for the entire city was 212 s, and the matching speed for the aerial photographs was considered practical.

The cover ratio of 3D point cloud data, defined as the ratio of point clouds to building surface, was calculated. Using the MMS points, a textured 3D city model was automatically generated for 46 buildings, with a cover ratio of more than 20% achieved successfully. The coverage calculation for the MMS point cloud required 92.5 s, whereas wall ID estimation and wall image creation required 845 s, with a total of 16 min for the two tasks.

## 6. Conclusions

This study aimed to effectively utilize various types of building measurement data from different sources by developing an efficient matching method for various measurement data on buildings using mainly geometric properties and implementing it as a Web API.

Specifically, for the 2D building polygons, we defined the overlap ratio between the 2D polygons and applied it to 88,525 residential buildings in Numazu City. Moreover, the developed 2D matching method was applied to aerial images to perform super-resolution processing, building extraction processing, and matching of the extracted building polygons. Although the number of buildings extracted from the aerial photographs was only 70.9% of the total number of PLATEAU buildings, 93.6% of the extracted buildings matched the PLATEAU building IDs, which holds significant practical potential.

**Table 4. Summary of the matching results of the various measurement data used in this study.**

|  | Data type and size | Dimension | Time (*2) | Qualitative Evaluation | Note |
|---|---|---|---|---|---|
| 2D Footprint (residential map) | GPKG/69.9 MB (All Numazu city) | 2D | 104 s | Classifying 1.2% as polygons requiring visual judgment | Visual confirmation of suggested polygons is required |
| 2D Footprint (extracted from aerial images) | GPKG/138.8 MB (All Numazu city) | 2D | 212 s (*3) | Classifying 6.4% as polygons requiring visual judgment (*4) | Visual confirmation of suggested or rejected polygons is required |
| MMS Point Cloud | LAS/3.2 GB (*1) (1 mesh) | 3D | 16 min | Cover ratio is mostly over 20% | 46 textured building files were made |
| LP Point Cloud | LAS/201.4 MB (*1) (1 mesh) | 3D | (Not measured) | Cover ratio is mostly over 40% | The wall image was mostly gray |

(*1) The mesh size was 400 m x 300 m, and the MMS point cloud measurement area was confined to a single prefectural road. (*2) Matching was performed using an AWS EC2 r5a.large environment, which had 16 GB memory and 2vCPUs. (*3) Preprocessing (building extraction) necessitated 3 h and 24 min in a GPU environment (4 Nvidia A100 GPUs). (*4) During the preprocessing step for matching, the number of buildings extracted from the aerial images was only 70.9% of the total number of PLATEAU buildings.

The 3D point cloud data were linked to 3D geometries such as PLATEAU LOD1 and LOD2, and a textured 3D city model was automatically created for 46 buildings. Although the processing time was longer than that for 2D data, the cover ratio, defined as the percentage of point clouds on the building surface, was generally over 20% for MMS measured along roads and over 40% for LP measured from the sky. This clarified the coverage status depending on the measurement method.

In this study, 3D point cloud data and aerial photographs, which are generally available as open data from national and local governments, were matched to a 3D building model. In contrast to GSV, which was widely used to create building textures in previous studies, the building texture information source utilized in this study was open data. Therefore, a 3D urban model with textures created through this approach could be released to the public and used in various fields in the future. Consequently, the textured 3D city model created by this method could be publicly released and utilized in numerous fields.

Finally, we discuss the limitations of this study. One limitation of matching 2D polygons is that it handles cases in which polygons representing the same building are initially parallel-shifted and have a small or nonexistent overlap. In the proposed method, the suggestion process linked polygons if the distance between the centers of gravity was less than 10 m and the difference in area was less than 20%. However, in the case of large parallel shift distances, the criterion of a distance between the centers of gravity of 10 m or less was not met.

Aerial images provide valuable information on the roofs of buildings, which is difficult to obtain from ground-based observations. Based on the results of the 2D polygon matching proposed in this study, ways to increase the information on roofs in 3D city models must be investigated.

One limitation of point-cloud data matching is the use of LP point clouds. Although the LP point cloud has the advantage of high coverage compared to the MMS point cloud, ways to use the public LP point cloud to enhance the information of the 3D urban model must be considered.

## Acknowledgments

We would like to express our gratitude to Mr. Omata from the Center for Spatial Information Science, University of Tokyo, and Mr. Endo from the Association for Promotion of Infrastructure Geospatial Information Distribution for their invaluable assistance in analyzing the data in this study.

## Author Contributions

**Conceptualization:** Yoshiki Ogawa, Yoshihide Sekimoto.

**Data curation:** Yoshiki Ogawa, Go Sato.

**Formal analysis:** Yoshiki Ogawa, Go Sato.

**Funding acquisition:** Yoshiki Ogawa, Yoshihide Sekimoto.

**Investigation:** Yoshiki Ogawa, Go Sato, Yoshihide Sekimoto.

**Methodology:** Yoshiki Ogawa, Go Sato, Yoshihide Sekimoto.

**Project administration:** Yoshiki Ogawa, Yoshihide Sekimoto.

**Resources:** Yoshiki Ogawa, Go Sato.

**Software:** Yoshiki Ogawa, Go Sato.

**Supervision:** Yoshiki Ogawa, Yoshihide Sekimoto.

**Validation:** Yoshiki Ogawa, Go Sato.

**Visualization:** Yoshiki Ogawa, Go Sato, Yoshihide Sekimoto.

**Writing – original draft:** Go Sato.

**Writing – review & editing:** Yoshiki Ogawa, Yoshihide Sekimoto.

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
