## [Decision Letter · Decision Letter 0]

15 Aug 2023

PONE-D-23-16713Geometric-Based Approach for Linking Various Building Measurement Data to a 3D City ModelPLOS ONE

Dear Dr. Ogawa,

Thank you for submitting your manuscript to PLOS ONE. After careful consideration, we feel that it has merit but does not fully meet PLOS ONE’s publication criteria as it currently stands. Therefore, we invite you to submit a revised version of the manuscript that addresses the points raised during the review process.

Please revise the article as per reviewer's comments specially related to methodology and research context, also during revision refer to relevant references only.

Please submit your revised manuscript by Sep 29 2023 11:59PM. If you will need more time than this to complete your revisions, please reply to this message or contact the journal office at plosone@plos.org. Please include the following items when submitting your revised manuscript:A rebuttal letter that responds to each point raised by the academic editor and reviewer(s). You should upload this letter as a separate file labeled 'Response to Reviewers'.A marked-up copy of your manuscript that highlights changes made to the original version. You should upload this as a separate file labeled 'Revised Manuscript with Track Changes'.An unmarked version of your revised paper without tracked changes. You should upload this as a separate file labeled 'Manuscript'.

We look forward to receiving your revised manuscript.

Kind regards,

Ayesha Maqbool, PhD

Academic Editor

PLOS ONE

4. We note that Figures 7,10 and 11 in your submission contain copyrighted images. All PLOS content is published under the Creative Commons Attribution License (CC BY 4.0), which means that the manuscript, images, and Supporting Information files will be freely available online, and any third party is permitted to access, download, copy, distribute, and use these materials in any way, even commercially, with proper attribution. For more information, see our copyright guidelines: http://journals.plos.org/plosone/s/licenses-and-copyright.

a. You may seek permission from the original copyright holder of Figures 1,2,5,6,8 and 9 to publish the content specifically under the CC BY 4.0 license.

5. We note that Figures 1,2,5,6,8 and 9 in your submission contain [map/satellite] images which may be copyrighted. All PLOS content is published under the Creative Commons Attribution License (CC BY 4.0), which means that the manuscript, images, and Supporting Information files will be freely available online, and any third party is permitted to access, download, copy, distribute, and use these materials in any way, even commercially, with proper attribution. For these reasons, we cannot publish previously copyrighted maps or satellite images created using proprietary data, such as Google software (Google Maps, Street View, and Earth). For more information, see our copyright guidelines: http://journals.plos.org/plosone/s/licenses-and-copyright.

a. You may seek permission from the original copyright holder of Figures 1,2,5,6,8 and 9 to publish the content specifically under the CC BY 4.0 license. 

Reviewers' comments:

Reviewer's Responses to Questions

**Comments to the Author**

1. Is the manuscript technically sound, and do the data support the conclusions?

Reviewer #1: Partly

Reviewer #2: Yes

Reviewer #3: No

2. Has the statistical analysis been performed appropriately and rigorously? 

Reviewer #1: I Don't Know

Reviewer #2: Yes

Reviewer #3: No

3. Have the authors made all data underlying the findings in their manuscript fully available?

Reviewer #1: Yes

Reviewer #2: Yes

Reviewer #3: No

4. Is the manuscript presented in an intelligible fashion and written in standard English?

Reviewer #1: Yes

Reviewer #2: Yes

Reviewer #3: No

5. Review Comments to the Author

Reviewer #1: This paper is intended to combine different types of data to develop a 3D city model. In today's cities, the concept of smart cities is clearly in need of multi-layer digital models of urban areas, and contributions like this one are always welcome.

Being aware of the limitations of their work, the authors achieved a high matching rate, which reveals the suitability of their method.

However, there are a few changes to be made to improve the quality of the paper:

- Abstract (and throughout the paper)

Please specify that the Ministry is from Japan.

The source of the point cloud data (aerial surveying or vehicle-mounted LiDAR) should be indicated the first time they are mentioned.

What do the Authors mean by "a private map"? In this field, one expects to see government or institutional maps.

- Introduction

Please expand the research context at the beginning of this section. After that, the PLATEAU project and this paper's focus can be addressed.

Acronyms should be fully written the first time they appear in the paper. E.g., MMS

- Methodology

Given that a web API was developed, it is necessary to show some code, maybe in the form of simplified scripts, of the processes computed.

- Related work

At the end of this section, in view of previous research on the topic, the research gap should be synthesised, and the need of this paper justified.

- Conclusions

This section should not literally replicate the results data but constitute a synthesis of the research outcomes and their implications.

Reviewer #2: Dear authors,

I have read carefully your manuscript entitled "Geometric-Based Approach for Linking Various Building Measurement Data to a 3D City Model". I consider this paper could be accepted for publication as the results are promising. Nevertheless, before its final acceptance, some issues should be undertaken.

My main concern is on references and their format:

-The citation referring to the Ministry of Land, Infrastructure, Transport and Tourism is not consistently written in the text (a word is missing twice).

-There are two references written in capital letters.

-Finally, in my view, authors should include some recent references on the projective transformation usage when dealing with point clouds or walls (or façades). Please, consider including https://doi.org/10.3390/s23156924 and https://doi.org/10.3390/sym13020335

Kind regards and congratulations on your research.

Reviewer #3: Summary

The study presents a Web API application for matching 2D polygons representing building footprints from different sources and a 3D point cloud with the 3D building models to create wall textures. The tests are carried out for the open 3D city model known as PLATEAU.

Overall comments

In my opinion, this paper can’t be published due to low presentation and the lack of novelty. I recommend that this paper be rejected. The paper is written like a general description, no scientific detail about the applied methods being presented.

In the abstract the authors mentioned different building measurement data from different sources, but throughout the paper only buildings footprints represented as polygons, 3D point clouds and buildings 3D models are mentioned. By the word “measurement”, the reader may be thinking about building height or other numeric information related to the building obtained by using different measurement instruments. Moreover, by “3D point clouds matching”, the reader can understand that two different point clouds has to be matched, as by “2D footprint matching”, two polygons are matched, but here, a 3D point cloud is matched with a building 3D model.

The method that projects the 3D point clouds on buildings facades to create textured walls, was published before. Please see:

Oniga, E.: A NEW APPROACH FOR THE SEMI-AUTOMATIC TEXTURE GENERATION OF THE BUILDINGS FACADES, FROM TERRESTRIAL LASER SCANNER DATA, Int. Arch. Photogramm. Remote Sens. Spatial Inf. Sci., XXXIX-B6, 161–166, https://doi.org/10.5194/isprsarchives-XXXIX-B6-161-2012, 2012.

Specific comments

1. “aerial laser point cloud” should be reformulate as “Airborne Laser Scanner (ALS) point cloud”.

2. Figure 1 is of poor quality.

3. Figure 3 (a2), a scale bar should be added as the threshold for the polygons centroids matching is 10 m.

4. “aero images” reformulate to “aerial images”.

5. “aircraft data” reformulate to “airborne data”.

6. “images created from oblique images”? The images are created from images??

7. Explain the meaning of the MMS acronym.

6. PLOS authors have the option to publish the peer review history of their article (what does this mean?). If published, this will include your full peer review and any attached files.

Reviewer #1: No

Reviewer #2: No

Reviewer #3: No

---

## [Author Response · Author response to Decision Letter 0]

1 Dec 2023

Manuscript PONE-D-23-16713

Response to Reviewers

Dear Editor and Reviewers,

We are grateful for the insightful comments and criticisms of our paper. We have incorporated all the reviewers' suggestions in the manuscript's new version. We hope that you will now find it appropriate for a recommendation. Those changes are highlighted in yellow within the manuscript.

Best wishes,

Yoshiki Ogawa, on behalf of all authors

and

Author response: In response to the comment 1, we changed the file names.

Author response: In response to the comment 2 and 3, we added the explanation after the Abstract as below.

Data Availability: The image data and LP/ MMS data analyzed in the study are provided by the Geospatial Information Authority of Japan (https://www.gsi.go.jp/ENGLISH/index.html) and Shizuoka prefecture via Association for Promotion of Infrastructure Geospatial Information Distribution (https://front.geospatial.jp/). These are open data and can be shared publicly. Our code is shared via GitHub (https://github.com/Project-PLATEAU/UC22-008-Building-matching-WebAPI).

4. We note that Figures 7,10 and 11 in your submission contain copyrighted images. All PLOS content is published under the Creative Commons Attribution License (CC BY 4.0), which means that the manuscript, images, and Supporting Information files will be freely available online, and any third party is permitted to access, download, copy, distribute, and use these materials in any way, even commercially, with proper attribution. For more information, see our copyright guidelines: http://journals.plos.org/plosone/s/licenses-and-copyright.

a. You may seek permission from the original copyright holder of Figures 1,2,5,6,8 and 9 to publish the content specifically under the CC BY 4.0 license.

5. We note that Figures 1,2,5,6,8 and 9 in your submission contain [map/satellite] images which may be copyrighted. All PLOS content is published under the Creative Commons Attribution License (CC BY 4.0), which means that the manuscript, images, and Supporting Information files will be freely available online, and any third party is permitted to access, download, copy, distribute, and use these materials in any way, even commercially, with proper attribution. For these reasons, we cannot publish previously copyrighted maps or satellite images created using proprietary data, such as Google software (Google Maps, Street View, and Earth). For more information, see our copyright guidelines: http://journals.plos.org/plosone/s/licenses-and-copyright.

a. You may seek permission from the original copyright holder of Figures 1,2,5,6,8 and 9 to publish the content specifically under the CC BY 4.0 license. 

Author response: In response to the comment 4 and 5, we added the explanation for the figure 1, 2, 6, 7, 10 and 11. Since the data is open data under CCBY4.0, "VIRTUAL SHIZUOKA © Shizuoka prefecture (Licensed under CC BY 4.0)" was added for the point cloud. "Aerial photograph ©Geospatial Information Authority of Japan (Licensed under CC BY 4.0)" was added for the aerial photographs. We use completely open data aerial imagery and other imagery that is free of copyright, so unlike Google imagery or paid imagery, we do not need to ask permission.

Review Comments to the Author:

Reviewer #1: This paper is intended to combine different types of data to develop a 3D city model. In today's cities, the concept of smart cities is clearly in need of multi-layer digital models of urban areas, and contributions like this one are always welcome.

Being aware of the limitations of their work, the authors achieved a high matching rate, which reveals the suitability of their method.

However, there are a few changes to be made to improve the quality of the paper:

- Abstract (and throughout the paper)

Please specify that the Ministry is from Japan.

Author response: As suggested by the reviewer, we specified that the Ministry is from Japan.

The source of the point cloud data (aerial surveying or vehicle-mounted LiDAR) should be indicated the first time they are mentioned.

Author response: In response to the reviewer's comment, we specified that the source of the point cloud data was MMS.

What do the Authors mean by "a private map"? In this field, one expects to see government or institutional maps.

Author response: We have added the statement "maps created by a private company" because the relevant section is a description of a map created by a private company.

- Introduction

Please expand the research context at the beginning of this section. After that, the PLATEAU project and this paper's focus can be addressed.

Author response: We explained in 1. Introduction that 3D urban model development is not limited to PLATEAU but is taking place all over the world.

Acronyms should be fully written the first time they appear in the paper. E.g., MMS

Author response: In response to the reviewer's comment, we modified the usage of acronyms in 1. Introduction.

- Methodology

Given that a web API was developed, it is necessary to show some code, maybe in the form of simplified scripts, of the processes computed.

Author response: The URL of the website (https://project-plateau.github.io/UC22-008-Building-matching-WebAPI/) is provided in 4. Methodology, as the description of the relevant part is available on the website.

- Related work

At the end of this section, in view of previous research on the topic, the research gap should be synthesised, and the need of this paper justified.

Author response: At the end of the section, we point out improvements in this paper compared to existing methods.

- Conclusions

This section should not literally replicate the results data but constitute a synthesis of the research outcomes and their implications. 

Author response: The repetition of numbers in the 2D matching results was removed in 6. Conclusions as it was deemed redundant and unnecessary.

Reviewer #2: Dear authors,

I have read carefully your manuscript entitled "Geometric-Based Approach for Linking Various Building Measurement Data to a 3D City Model". I consider this paper could be accepted for publication as the results are promising. Nevertheless, before its final acceptance, some issues should be undertaken.

My main concern is on references and their format:

-The citation referring to the Ministry of Land, Infrastructure, Transport and Tourism is not consistently written in the text (a word is missing twice).

Author response: In response to the reviewer's comment, we modified the designated words in the citation.

-There are two references written in capital letters.

Author response: In response to the reviewer's comment, we corrected references written in capital letters to lower case.

-Finally, in my view, authors should include some recent references on the projective transformation usage when dealing with point clouds or walls (or façades). Please, consider including https://doi.org/10.3390/s23156924 and https://doi.org/10.3390/sym13020335.

Author response: As suggested by the reviewer, we cited https://doi.org/10.3390/s23156924 and https://doi.org/10.3390/sym13020335 in 2. Related work.

Kind regards and congratulations on your research.

Reviewer #3: Summary

The study presents a Web API application for matching 2D polygons representing building footprints from different sources and a 3D point cloud with the 3D building models to create wall textures. The tests are carried out for the open 3D city model known as PLATEAU.

Overall comments

In my opinion, this paper can’t be published due to low presentation and the lack of novelty. I recommend that this paper be rejected. The paper is written like a general description, no scientific detail about the applied methods being presented.

In the abstract the authors mentioned different building measurement data from different sources, but throughout the paper only buildings footprints represented as polygons, 3D point clouds and buildings 3D models are mentioned. By the word “measurement”, the reader may be thinking about building height or other numeric information related to the building obtained by using different measurement instruments. Moreover, by “3D point clouds matching”, the reader can understand that two different point clouds has to be matched, as by “2D footprint matching”, two polygons are matched, but here, a 3D point cloud is matched with a building 3D model.

The method that projects the 3D point clouds on buildings facades to create textured walls, was published before. Please see:

Oniga, E.: A NEW APPROACH FOR THE SEMI-AUTOMATIC TEXTURE GENERATION OF THE BUILDINGS FACADES, FROM TERRESTRIAL LASER SCANNER DATA, Int. Arch. Photogramm. Remote Sens. Spatial Inf. Sci., XXXIX-B6, 161–166, https://doi.org/10.5194/isprsarchives-XXXIX-B6-161-2012, 2012.

Author response: We cited https://doi.org/10.5194/isprsarchives-XXXIX-B6-161-2012, 2012 and explained the difference in 2. Related work as below.

E. Oniga (2012) proposed an algorithm for the semiautomatic texture generation based on color information, RGB values of every point captured by terrestrial laser scanning technology, and 3D surfaces defining building facades generated using commercial 3D software. The operator needed to define the limiting value, i.e., the minimum distance between a point and the closest surface. In this study, the threshold used to choose the points to be projected was set automatically.

Specific comments

1. “aerial laser point cloud” should be reformulate as “Airborne Laser Scanner (ALS) point cloud”.

Author response: In response to the reviewer's comment, we reformulated the designated words in 2. Related work.

2. Figure 1 is of poor quality.

Author response: Since the figure in question is a conceptual figure and a detailed explanation is provided in Fig. 2, it was determined that the figure itself does not need to be changed. The word "conceptual" was added to the title of the figure to make it easier to understand that this figure is a conceptual diagram.

3. Figure 3 (a2), a scale bar should be added as the threshold for the polygons centroids matching is 10 m.

Author response: Since the corresponding figure shows the geometrical relationship between two-dimensional polygons, it was determined that no additional scale was necessary.

4. “aero images” reformulate to “aerial images”.

Author response: As suggested by the reviewer, we reformulated the designated words.

5. “aircraft data” reformulate to “airborne data”.

Author response: As suggested by the reviewer, we reformulated the designated words.

6. “images created from oblique images”? The images are created from images??

Author response: The indicated sections were deleted in 2. Related work because the same content was continuously included.

7. Explain the meaning of the MMS acronym. 

Author response: The phrase Mobile Mapping System was added in 1. Introduction, where the word MMS first appears in the text.

---

## [Decision Letter · Decision Letter 1]

14 Dec 2023

Geometric-Based Approach for Linking Various Building Measurement Data to a 3D City Model

PONE-D-23-16713R1

Dear Dr. Ogawa,

We’re pleased to inform you that your manuscript has been judged scientifically suitable for publication and will be formally accepted for publication once it meets all outstanding technical requirements.

Kind regards,

Ayesha Maqbool, PhD

Academic Editor

PLOS ONE

Additional Editor Comments (optional):

Reviewers' comments:

Reviewer's Responses to Questions

**Comments to the Author**

1. If the authors have adequately addressed your comments raised in a previous round of review and you feel that this manuscript is now acceptable for publication, you may indicate that here to bypass the “Comments to the Author” section, enter your conflict of interest statement in the “Confidential to Editor” section, and submit your "Accept" recommendation.

Reviewer #1: All comments have been addressed

Reviewer #2: All comments have been addressed

2. Is the manuscript technically sound, and do the data support the conclusions?

Reviewer #1: Yes

Reviewer #2: Yes

3. Has the statistical analysis been performed appropriately and rigorously? 

Reviewer #1: Yes

Reviewer #2: Yes

4. Have the authors made all data underlying the findings in their manuscript fully available?

Reviewer #1: Yes

Reviewer #2: Yes

5. Is the manuscript presented in an intelligible fashion and written in standard English?

Reviewer #1: Yes

Reviewer #2: Yes

6. Review Comments to the Author

Reviewer #1: Authors addressed all my comments. Just a minor change:

- Authors specified that the source of the point cloud data was MMS, but please, do it in the Introduction instead of in the Abstract.

Reviewer #2: Dear authors,

Thank you for addressing the comments I suggested to you. In my view, the manuscript is now ready to be accepted.

Kind regards

7. PLOS authors have the option to publish the peer review history of their article (what does this mean?). If published, this will include your full peer review and any attached files.

Reviewer #1: No

Reviewer #2: No

---

## [Editor Report · Acceptance letter]

26 Dec 2023

PONE-D-23-16713R1 

PLOS ONE

Dear Dr. Ogawa, 

I'm pleased to inform you that your manuscript has been deemed suitable for publication in PLOS ONE. Congratulations! Your manuscript is now being handed over to our production team.

Kind regards, 

on behalf of

Dr. Ayesha Maqbool 

Academic Editor

PLOS ONE